# Light-activated Frizzled7 reveals a permissive role of non-canonical wnt signaling in mesendoderm cell migration

Daniel Čapek[1], Michael Smutny[1,2], Alexandra-Madelaine Tichy[3,4], Maurizio Morri[1†], Harald Janovjak[1,3,4], Carl-Philipp Heisenberg[1]*

[1]Institute of Science and Technology Austria, Klosterneuburg, Austria; [2]Centre for Mechanochemical Cell Biology and Division of Biomedical Sciences, Warwick Medical School, University of Warwick, Coventry, United Kingdom; [3]Australian Regenerative Medicine Institute (ARMI), Faculty of Medicine, Nursing and Health Sciences, Monash University, Clayton, Australia; [4]European Molecular Biology Laboratory Australia (EMBL Australia), Monash University, Clayton, Australia

*For correspondence:
heisenberg@ist.ac.at

Present address: [†]Chan Zuckerberg Biohub, San Francisco, United States

Competing interests: The authors declare that no competing interests exist.

**Abstract** Non-canonical Wnt signaling plays a central role for coordinated cell polarization and directed migration in metazoan development. While spatiotemporally restricted activation of non-canonical Wnt-signaling drives cell polarization in epithelial tissues, it remains unclear whether such instructive activity is also critical for directed mesenchymal cell migration. Here, we developed a light-activated version of the non-canonical Wnt receptor Frizzled 7 (Fz7) to analyze how restricted activation of non-canonical Wnt signaling affects directed anterior axial mesendoderm (prechordal plate, ppl) cell migration within the zebrafish gastrula. We found that Fz7 signaling is required for ppl cell protrusion formation and migration and that spatiotemporally restricted ectopic activation is capable of redirecting their migration. Finally, we show that uniform activation of Fz7 signaling in ppl cells fully rescues defective directed cell migration in *fz7* mutant embryos. Together, our findings reveal that in contrast to the situation in epithelial cells, non-canonical Wnt signaling functions permissively rather than instructively in directed mesenchymal cell migration during gastrulation.
DOI: https://doi.org/10.7554/eLife.42093.001

## Introduction

Non-canonical Wnt-Frizzled (Fz)/planar cell polarity (PCP) signaling plays a decisive role in cell polarization and coordinated tissue morphogenesis in metazoan development (*Gray et al., 2011*; *Singh and Mlodzik, 2012*). Initially, core components of the Wnt-Fz/PCP signaling module were identified in *Drosophila* to determine coordinated planar polarity of epithelial cells (*Gubb and García-Bellido, 1982*; *Lawrence and Shelton, 1975*; *Vinson and Adler, 1987*; *Seifert and Mlodzik, 2007*). Subsequently, vertebrate orthologs of key Wnt-Fz/PCP signaling components were shown to be required for both epithelial planar cell polarity and coordinated mesenchymal cell polarization, migration and intercalation (*Gray et al., 2011*; *Singh and Mlodzik, 2012*). Notably, signaling molecules and mechanisms determining epithelial cell polarity are largely conserved between vertebrate and invertebrate organisms indicating the general importance of Wnt-Fz/PCP in animal development (*Yang and Mlodzik, 2015*; *Seifert and Mlodzik, 2007*).

The key feature of Wnt-Fz/PCP signaling to achieve epithelial cell polarization is the polarized subcellular distribution of Wnt-Fz/PCP components in a tissue-wide and coordinated manner (*Strutt, 2001*; *Feiguin et al., 2001*; *Tree et al., 2002*). This polarized distribution and activation of Wnt-Fz/PCP components is thought to trigger spatially confined changes in the cytoskeletal

architecture of epithelial cells leading to their morphological recognizable polarization (*Strutt and Warrington, 2008*; *Ségalen et al., 2010*; *Strutt et al., 2011*; *Butler and Wallingford, 2018*). In contrast, much less is known about how Wnt-Fz/PCP signaling controls mesenchymal cell polarization and movement (*Shindo and Wallingford, 2014*).

Previous studies on non-canonical Wnt signaling in mesenchymal cells have demonstrated that Wnt-Fz/PCP signaling components with a known function in epithelial PCP are required for proper cell polarization and intercalation (*Gray et al., 2011*). Moreover, the Wnt-Fz/PCP core components Prickle (Pk) and Dishevelled (Dsh) have been shown to localize preferentially to the anterior or posterior site of mesenchymal cells, respectively, pointing to the possibility that localized activity of these components triggers mesenchymal cell polarization (*Ciruna et al., 2006*; *Yin et al., 2008*). Likewise, studies on the role of non-canonical Wnt signaling in mesenchymal myoblast cells provided evidence for the capacity of graded Wnt11 signals in directing myoblast cell elongation (*Gros et al., 2009*). These data suggest that non-canonical Wnt signaling instructively polarizes mesenchymal cells similar to its function in epithelial cells (*Figure 1A*). In contrast, the observation that uniform overexpression of non-canonical Wnt signaling ligands can rescue defective mesenchymal cell polarization in Wnt-Fz/PCP mutant embryos (*Heisenberg et al., 2000*) points to a permissive function of Wnt-Fz/PCP signaling in this process (*Figure 1A*). While these overexpression and localization studies suggest both a permissive and instructive role for Wnt-Fz/PCP signaling in mesenchymal cell polarization and intercalation, direct evidence for non-canonical Wnt signaling exerting either of these functions is still lacking.

In zebrafish, Wnt-Fz/PCP signaling is required for directed migration of mesenchymal mesendoderm progenitors during gastrulation. Specifically, migration directionality and coordination of anterior axial mesendoderm, the so-called prechordal plate (ppl) progenitors, is diminished in loss-of-function mutants for the Wnt-Fz/PCP ligand *wnt11* (*Ulrich et al., 2003*; *Ulrich et al., 2005*). Ppl progenitors are specified at the dorsal germ ring margin at the onset of gastrulation (*Warga and Kimmel, 1990*; *Kimmel et al., 1995*; *Montero et al., 2005*). Upon specification, ppl progenitors internalize by coordinated ingression and then migrate as a coherent cluster of cells in a straight path from the germ ring margin towards the animal pole of the gastrula (*Warga and Kimmel, 1990*; *Montero et al., 2005*). Recent work showed that migration of ppl progenitors between the yolk cell and the overlying ectoderm can generate frictional forces, which modulate ectoderm cell movements during gastrulation (*Smutny et al., 2017*).

We reasoned that instructive and permissive functions of non-canonical Wnt signaling in mesenchymal cell polarization and migration could be distinguished using an approach that allows uniform activation of Wnt-Fz/PCP signaling within the entire cell population. If the ppl was indeed guided by an instructive mechanism, uniform activation should not be able to rescue the ppl cell migration defect in Wnt-Fz/PCP signaling defective embryos. Uniformly activating Wnt-Fz/PCP signaling requires cells to be non-responsive to endogenous Wnt ligands. To achieve ligand-independent uniform activation, we have developed a light-activated version of the Wnt-Fz/PCP receptor Fz7, allowing us to trigger Fz7/PCP signaling in a spatiotemporally controlled manner directly at the receptor level. Using this Opto-Fz7 receptor in *fz7* mutant embryos, we show that localized Fz7 signaling has the potential to direct ppl cell migration. Unexpectedly, however, uniform activation of Fz7 signaling was also sufficient to rescue defective ppl cell migration in *fz7* mutant embryos. This suggests that endogenous Fz7/PCP signaling has a permissive rather than instructive function in directed ppl cell migration during gastrulation.

## Results

For analyzing how Wnt-Fz/PCP signaling functions in directed mesenchymal cell migration, we turned to zebrafish embryos, which have previously been shown to require Wnt-Fz/PCP signaling for directed migration of mesendoderm progenitor cells during gastrulation (*Topczewski et al., 2001*; *Jessen et al., 2002*; *Ulrich et al., 2003*; *Dumortier et al., 2012*). Specifically, we determined how directed migration of ppl progenitors is affected in mutant embryos for the Wnt11 receptor Fz7 (*Djiane et al., 2000*; *Witzel et al., 2006*), which display strongly impaired gastrulation movements (*Quesada-Hernández et al., 2010*). Our analysis showed that in MZ*fz7a/b* double mutants, beside the previously reported dorsal convergence defects (*Quesada-Hernández et al., 2010*) (*Figure 1—figure supplement 1*, *Figure 1—source data 1*), the ppl was elongated and placed closer to the

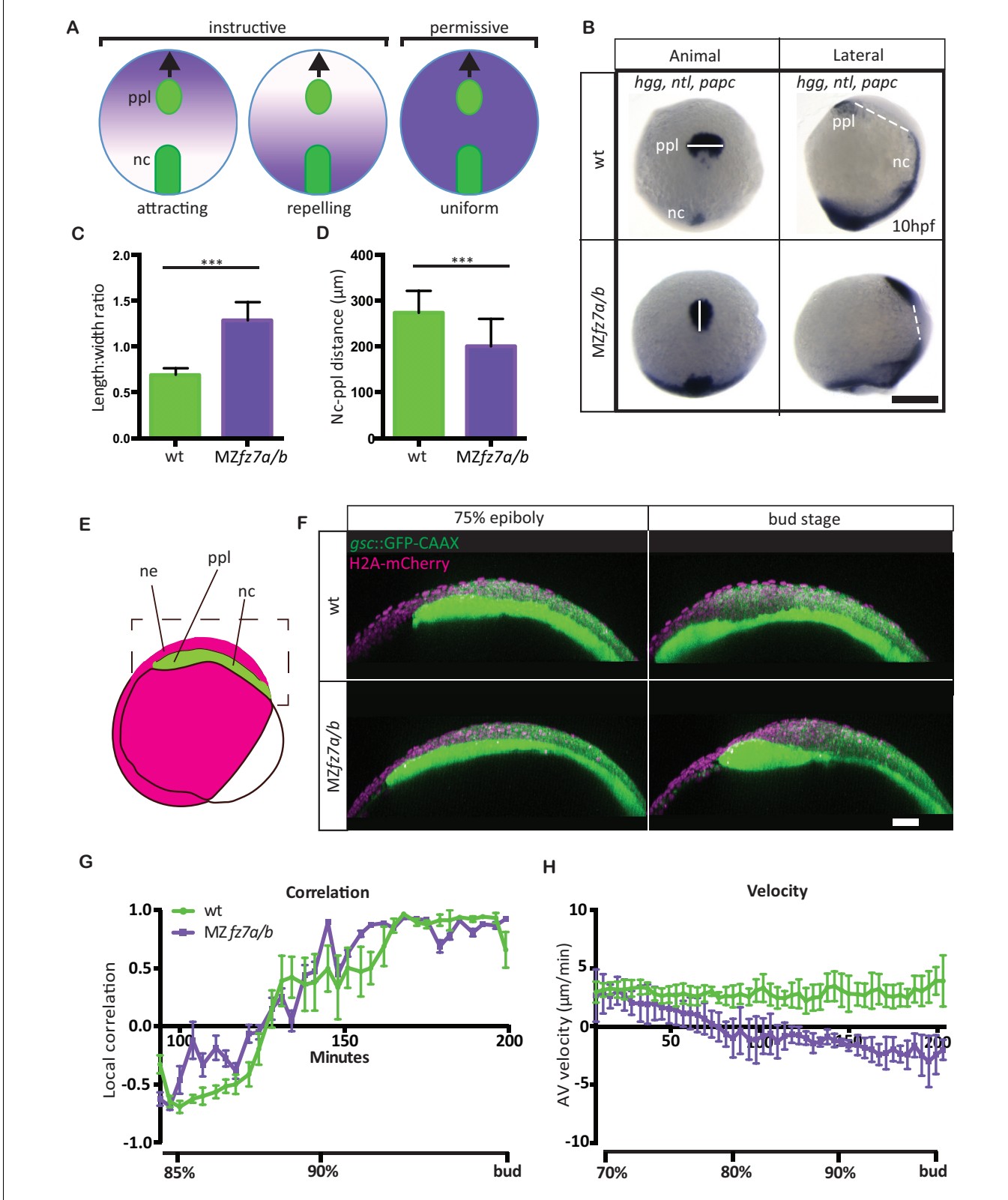

**Figure 1.** MZ*fz7a/b* prechordal plate phenotype. (**A**) Possible modes of Wnt-Fz/PCP function in ppl migration. The direction of ppl (green) migration (black arrows) could be influenced by an attracting or repellent gradient of Wnt-Fz/PCP signaling (purple), which polarizes ppl cells, and thus directs their migration instructively. Alternatively, uniform Wnt-Fz/PCP signaling could operate in a permissive mode, allowing ppl cells to undergo directed migration by for example controlling general cell motility. (**B**) Animal (left column) and lateral (right column) views of the notochord (nc) and prechordal

*Figure 1 continued on next page*

*Figure 1 continued*

plate (ppl) in wt (upper row) and MZ*fz7a/b* mutant (lower row) embryos at the end of gastrulation (bud stage, 10hpf) labeled by in situ hybridization for *hgg* (ppl), *ntl* (nc), and *papc* (paraxial mesoderm). Scale bar, 250 µm. (C,D) Length-to-width ratio of the ppl (C) and distance between the anterior end of the nc and the posterior end of the ppl (D) in wt (green) and MZ*fz7a/b* mutant (purple) embryos. Error bars are standard deviations; N = 23 (wt), 17 (MZ*fz7a/b*), ***p<0.001 (Student's t-test). (E) Schematic of embryo orientation used for imaging ppl and overlying neurectoderm (ne) movements in (F). Neurectoderm, magenta; ppl and nc, green; boxed area corresponds to the imaged area in (F). (F) Confocal microscopy images of wt (upper row) and MZ*fz7a/b* mutant (lower row) embryos at mid (left, 75% epiboly) and late (right, bud stage) gastrulation. Nuclei are marked by H2A-mCherry expression, and ppl and nc are marked by *gsc*:GFP-CAAX expression. Scale bar, 50 µm. (G) Local correlation between leading edge ppl cells and overlying ectodermal cells in wt (green) and MZ*fz7a/b* mutant (purple) embryos. X-axis, time (min, stage); y-axis, order parameter of local correlation; error bars are standard error of mean; N = 3 per genotype. (H) Velocities of ppl leading edge cells in wt (green) and MZ*fz7a/b* mutant (purple) embryos. X-axis, time (min, stage); y-axis, velocity in vegetal to animal pole direction; error bars are standard error of mean; N = 3 per genotype.
DOI: https://doi.org/10.7554/eLife.42093.002

The following source data and figure supplement are available for figure 1:

**Source data 1.** Characterization of MZ*fz7a/b* mutants at the embryo and tissue level.
DOI: https://doi.org/10.7554/eLife.42093.004
**Figure supplement 1.** MZ*fz7a/b* convergence and extension phenotypes.
DOI: https://doi.org/10.7554/eLife.42093.003

anterior end of the notochord at the end of gastrulation (bud stage, 10 hrs post-fertilization, hpf) (*Figure 1B–D*, *Figure 1—source data 1*). This phenotype points to the possibility that Fz7 is required for directed ppl cell migration in zebrafish.

To investigate this possibility, we determined how the movement velocity of ppl progenitors towards the animal pole of the gastrula and their movement coordination with the overlying ectoderm are affected in MZ*fz7a/b* mutants compared to wild type (wt) embryos. Consistent with previous observations (*Smutny et al., 2017*), we found that wt ppl progenitors migrated with a velocity of 2–3 µm/min towards the animal pole, and that at late gastrulation (90% epiboly) this movement was highly correlated with the movement of overlying ectoderm progenitors (*Figure 1G&H*, *Video 1*, *Figure 1—source data 1*). In comparison, ppl progenitors in MZ*fz7a/b* mutant embryos slowed down their animal pole directed movement during gastrulation, and eventually even moved backwards towards the vegetal pole from mid-gastrulation on (*Figure 1F&H*, *Video 1*, *Figure 1—source data 1*). Interestingly, movement coordination between ppl and ectoderm progenitors remained unchanged in MZ*fz7a/b* mutant embryos (*Figure 1G*, *Figure 1—source data 1*), arguing against a previously suggested function of Fz7 in controlling the interaction between ppl and overlying ectoderm (*Winklbauer et al., 2001*; *Montero et al., 2005*).

To test whether Fz7 is cell-autonomously required for ppl progenitor cell migration, we transplanted ppl cells between wt and MZ*fz7a/b* mutant embryos at the onset of gastrulation (shield stage, 6 hpf). As controls, we performed homotypic transplantations from wt donor into wt host and from mutant donor into mutant host embryos. When wt cells were transplanted into the ppl of MZ*fz7a/b* mutant embryos, these cells typically moved to the leading edge of the ppl and showed lower movement correlation with their neighboring host cells when compared to transplanted mutant cells in mutant embryos (*Figure 2A&D*, *Video 2*, *Figure 2—source data 1*). Conversely, mutant ppl progenitors transplanted into the ppl of wt embryos typically ended up closer to the posterior end of the ppl

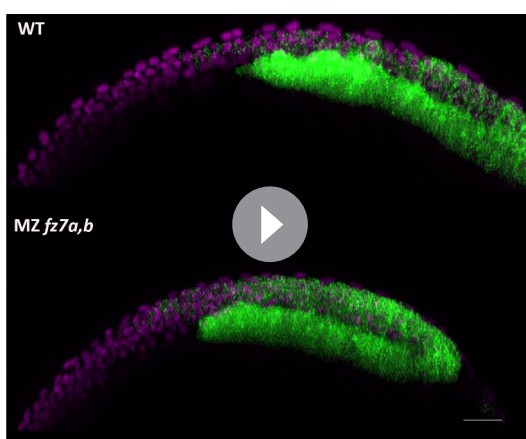

**Video 1.** Ppl and overlying neurectoderm cell movements in MZ*fz7a/b* mutant and wt Tg(*gsc*:GFP-CAAX) embryos from 7 to 10 hpf. Ppl cells are marked by *gsc*:GFP-CAAX expression (green) and all nuclei are stained by H2A-mCherry (magenta). Lateral view with animal pole to the left. Arrows point at ppl leading edge at the time the mutant ppl stops moving forward. Scale bar, 50 µm.
DOI: https://doi.org/10.7554/eLife.42093.005

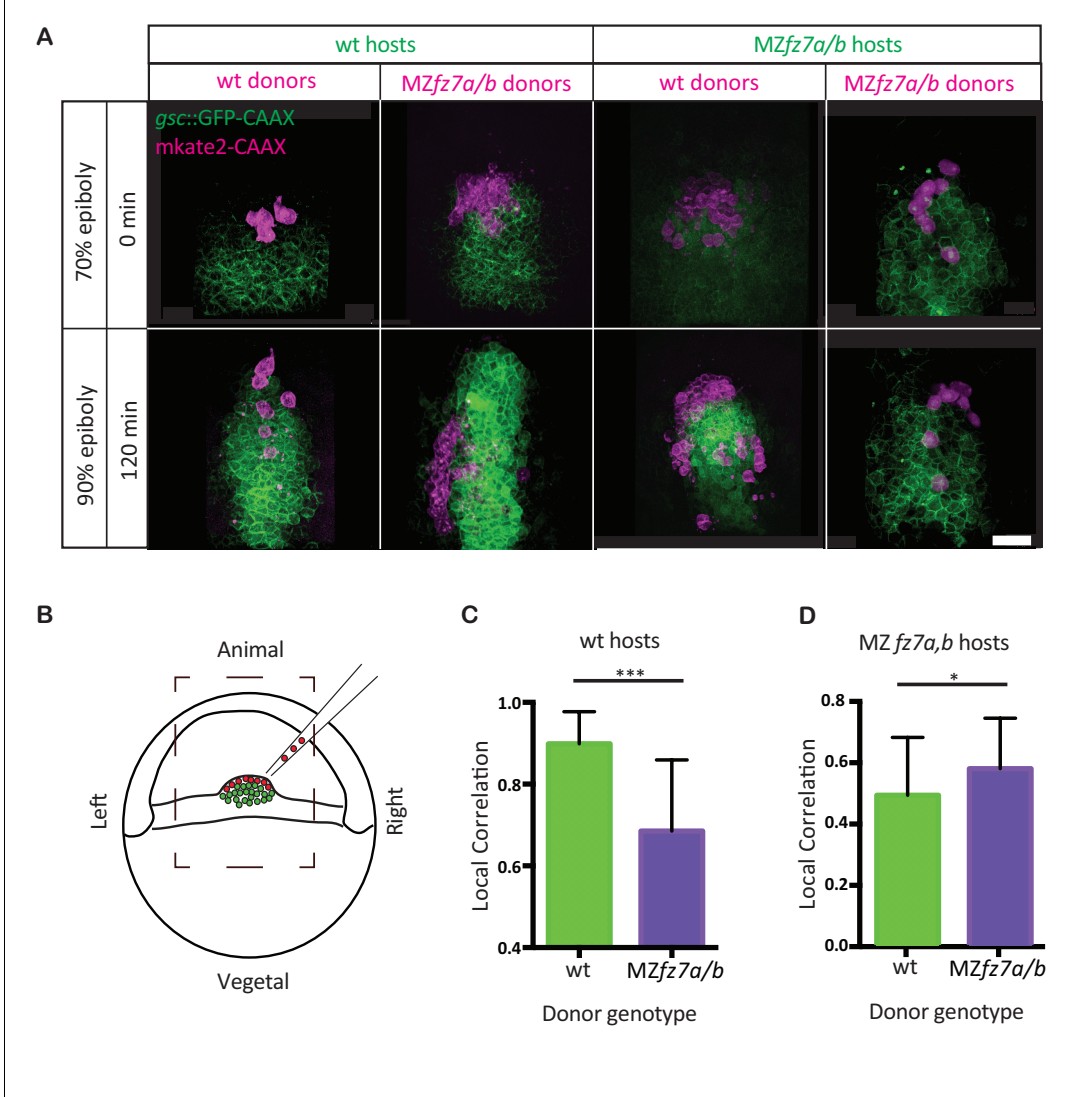

**Figure 2.** MZ*fz7a/b* prechordal plate progenitor migration. (**A**) 2-photon microscopy images of the transplanted ppl at early (70% epiboly; upper row) and late (90% epiboly, lower row) gastrulation. Dorsal views with animal to the top. Donor cells are marked by mkate2-CAAX expression (magenta) and host cells are marked by *gsc*:GFP-CAAX expression (green). Scale bar, 50 μm. (**B**) Schematic of the experimental setup of the transplantation experiments in (**A**). Cells from donor embryos consisting of ppl progenitors cells (magenta) were transplanted to the leading edge of the ppl (green) of a Tg(*gsc:GFP-CAAX*) or MZ*fz7a/b*; Tg(*gsc:GFP-CAAX*) host embryo. Boxed area outlines the field of view in (**A**). (**C,D**) Local correlation between host and donor cells within the transplanted ppl for wt (**C**) and MZ*fz7a/b* mutant (**D**) hosts. Error bars are standard deviations; N = 3 embryos for wt and MZ*fz7a/b* hosts each, ***p<0.001 and *p<0.05 (Student's t-test).

DOI: https://doi.org/10.7554/eLife.42093.006

The following source data is available for figure 2:

**Source data 1.** Characterization of MZ*fz7a/b* mutants cells in homotypic and heterotypic transplantation experiments.

DOI: https://doi.org/10.7554/eLife.42093.007

and showed reduced movement correlation with their wt neighboring cells when compared to transplanted wt cells in wt embryos (*Figure 2A&C*, *Video 2*, *Figure 2—source data 1*). Together, these transplantation experiments suggest that Fz7 functions cell-autonomously in ppl progenitors by determining their ability for undergoing directed migration.

To identify the cellular effector processes by which signaling through the Fz7 receptor affects directed ppl cell migration, we first took a reductionist approach and cultured isolated ppl progenitor cells in small clusters on Fibronectin-coated substrates and analyzed their protrusive behavior in

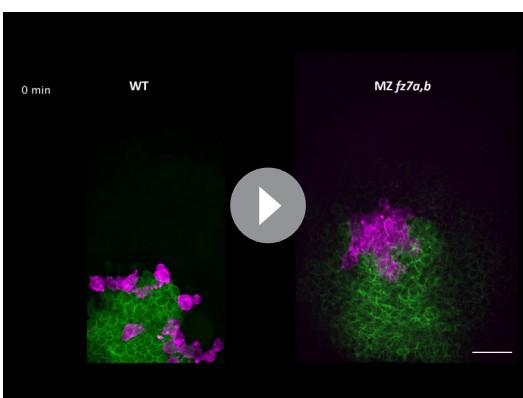

**Video 2.** Movement of MZ*fz7a/b* mutant and wt donor ppl cells (magenta) transplanted into the ppl of wt Tg (*gsc:GFP-CAAX*) host embryos. Transplanted cells are marked by mkate2-CAAX expression (magenta) and donor cells express *gsc*:GFP-CAAX (green). Scale bar, 50 µm.

DOI: https://doi.org/10.7554/eLife.42093.008

vitro. We found that after 4 hrs in culture wt progenitors at the margin of these clusters showed pronounced substrate spreading and formed multiple protrusions, resembling lamellipodia (54% of cells on the margin of clusters; *Figure 3A&B*). Ppl progenitors from MZ*fz7a/b* mutant embryos, in contrast, formed significantly less lamellipodia-like protrusions (27% of marginal cells; *Figure 3A&B*). This suggests that Fz7 is required for the formation of lamellipodia in ppl progenitors during substrate spreading. To assess the relevance of this in vitro observation for the situation in vivo, we compared protrusion formation of ppl cells in wt and MZ*fz7a/b* mutant embryos between 75% and 90% epiboly stage (8–9 hpf). Consistent with our in vitro observations, ppl cells formed significantly less protrusions resembling lamellipodia or pseudopodia in MZ*fz7a/b* mutants (0.14 ± 0.059 protrusions/cell/minute) compared to wt embryos (0.33 ± 0.075 protrusions/cell/minute), while the number of blebs increased and of protrusions resembling filopodia remained largely unchanged (*Figure 3C–F*, *Video 3*). Importantly, the preferential orientation of these different protrusion types in the direction of ppl progenitor cell migration towards the animal pole was indistinguishable between MZ*fz7a/b* and wt embryos (*Figure 3G*, *Figure 3—source data 1*). Together, this suggests that Fz7 is required for the type but not orientation of protrusions formed in leading edge ppl cells.

To investigate whether spatially restricted activation of Fz7 signaling has the potential to drive ppl cell polarization and direct their migration, we developed a light-activated version of Fz7. The rationale underlying our protein engineering approach was that the functional domains of seven-pass transmembrane (TM) receptors can be delineated in their canonical topology (*Airan et al., 2009*; *Siuda et al., 2015*; *Li et al., 2015*; *Gunaydin et al., 2014*; *Barish et al., 2013*; *van Wyk et al., 2015*; *Morri et al., 2018*). In particular, the N-terminus, three extracellular loops and seven TM domains are typically associated with ligand binding and receptor activation, whereas the three intracellular loops (ICL; ICL 1–3) and the C-terminus are typically associated with downstream signal transmission. Following this rationale, we substituted the intracellular domains of the light-sensitive receptor *rhodopsin* with the corresponding domains of *fz7* (*Figure 4A*, *Figure 4—figure supplement 1*). We identified these domains in *rhodopsin* and *fz7* sequences based on previous domain shuffling experiments of Class A and Class F GPCRs and mutagenesis analysis of *frizzled* receptors (*Liu et al., 2001*; *Kim et al., 2005*; *Tauriello et al., 2012*). We generated two receptor variants (*Figure 4A*, *Figure 4—figure supplement 1*). In the first variant, all intracellular elements of Rhodopsin were substituted by those of Fz7. In the second variant, taking into account that all known Dsh-binding domains of Fz are found in ICL3 and the C-terminus (*Wong et al., 2003*; *Schulte and Bryja, 2007*; *Tauriello et al., 2012*), the ICL1 and ICL2 of *rhodopsin* were retained to minimize perturbations to the original Rhodopsin fold. Because chimeric receptors of Rhodopsin and members of the more distantly related Class F GPCRs have never been generated, we applied molecular modeling to evaluate receptor compatibility. We found that the inactive and active state conformations of Rhodopsin and Smoothened (Smo), a surrogate receptor for Fz7 and the sole Class F GPCR for which inactive and active state structures are available, can be superimposed revealing a conserved arrangement of the TM helices (*Figure 4—figure supplement 2*; RMSD 2.8 and 2.6 Å for the inactive and active states, respectively). In addition, analysis of non-covalent contact networks (*Venkatakrishnan et al., 2016*; *Kayikci et al., 2018*) revealed similar numbers of inter-helical contacts that stabilize these receptors again in both receptor states (inactive state: 223 and 217 contacts for Rhodopsin and Smo; active state: 197 and 173 contacts for Rhodopsin and Smo). These contacts were distributed in a similar manner with ~70% of contacts occurring between helices TM1 and TM2, TM2 and TM3, TM3 and TM4/5, TM5 and TM6, and TM6 and TM7 in both receptors. Collectively,

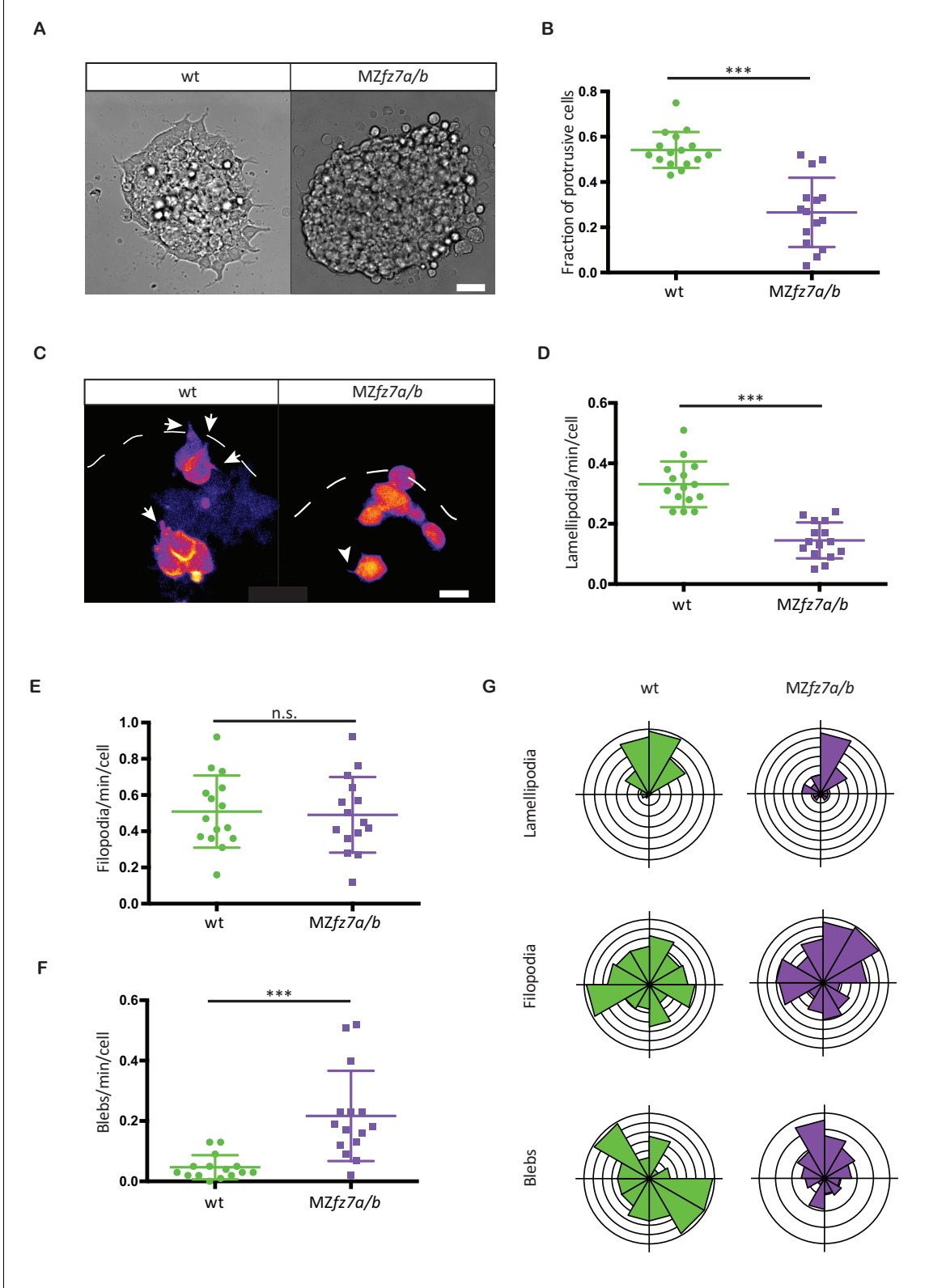

**Figure 3.** MZ*fz7a/b* prechordal plate progenitor protrusion formation. (**A**) Bright-field images of explants of wt (left) and MZ*fz7a/b* mutant (right) ppl tissue. Scale bar, 40 µm. (**B**) Fraction of cells at the explant margin displaying lamellipodia-like protrusions after 4 hrs in culture. Scatter plot with mean and standard deviation. N = 12 explants per genotype; ***p<0.001 (Mann-Whitney test). (**C**) Confocal images of transplanted wt donor ppl cells in wt host embryos (left) and MZ*fz7a/b* donor cells in MZ*fz7a/b* mutant host embryos (right). Transplanted cells are marked by mkate2-CAAX

*Figure 3 continued on next page*

*Figure 3 continued*

expression. Dashed white line marks the anterior edge of the ppl. Arrows point at lamellipodia-like protrusions, arrowheads at filopodia. Scale bar, 20 µm. (**D–F**) Number (calculated per cell and minute) of different types of protrusions in the transplanted cells shown in (**C**). Scatter plot with mean and standard deviation. N = 15 cells from five embryos per genotype. ***p<0.001, n.s., non significant (Student's t-test). (**G**) Orientation of different protrusion types in the transplanted cells shown in (**C**) (rose plots with animal top and vegetal bottom).
DOI: https://doi.org/10.7554/eLife.42093.009

The following source data is available for figure 3:

**Source data 1.** Protrusion formation in wt and MZ*fz7a/b* mutant embryos.
DOI: https://doi.org/10.7554/eLife.42093.010

these results suggest structural similarities toward the design of chimeric proteins that are non-responsive to endogenous ligands and, instead, can be activated by light.

To test whether these receptors can be used for spatiotemporally controlled ligand-independent activation of Fz7 signaling within the ppl, we first determined the subcellular localization of the two receptor-variants within progenitor cells. Following ectopic expression by mRNA injection at the one-cell stage, the second receptor variant containing the ICL3 and C-terminal intracellular domains of Fz7 showed clear and uniform localization at the plasma membrane of all progenitor cell types analyzed within the gastrulating embryo at 70% epiboly stage, including ppl, epiblast, nc and para-xial mesendoderm cells (7 hpf; *Figure 4B*, *Figure 4—figure supplement 3A–C*, *Figure 4—source data 1*). The first variant containing all four intracellular domains of Fz7, in contrast, did not clearly localize to the plasma membrane (*Figure 4—figure supplement 1B*), and we thus decided to solely focus on the second variant (termed Opto-Fz7) for the remainder of our study.

Next, we asked whether Opto-Fz7 is internalized upon light activation, given previous observations that non-canonical Fz receptor activation is followed by receptor internalization (*Onishi et al., 2013*). We found Opto-Fz7 to re-localize from the plasma membrane to intracellular vesicles upon light exposure (*Figure 4C*, *Figure 4—figure supplement 3A*, *Figure 4—source data 1*), suggesting that Opto-Fz7 signaling is activated by light. Next, we tested whether OptoFz7 specifically triggers non-canonical Wnt signaling, as suggested for Fz7 in zebrafish (*Quesada-Hernández et al., 2010*). To this end, we analyzed how the subcellular distrubution of Dvl and β-catenin change upon Opto-Fz7 activation, previously suggested to be sensitive to non-canonical (Dvl) and canonical Wnt signaling (β-catenin), respectively (*Park et al., 2005*; *Witzel et al., 2006*), *Yu et al., 2007*, *Kim et al., 2008a*; *Gao and Chen, 2010*). We found that zebrafish Dvl2-GFP co-localized with Opto-Fz7-mCherry at the plasma membrane, and that Dvl, much like Opto-Fz7, was endocytosed upon light activation (*Figure 4—figure supplement 4A&B*, *Figure 1—source data 1*). In contrast, we detected no recognizable changes in nuclear β-catenin localization upon light activation of Opto-Fz7 in embryos at 5.5 hpf, while uniform overexpression of the canonical Wnt ligand Wnt8 elicited nuclear accumulation of β-catenin throughout the embryo (*Figure 4—figure supplement 4C–F*, *Figure 4 –* source data; *Brunet et al., 2013*). Together, these findings indicate that Opto-Fz7 can be used for spatio-temporally controlled activation of non-canonical Fz7 signaling within the ppl.

Next, we asked whether spatially restricted activation of Fz7 signaling could affect ppl pro-genitor cell migration. To this end, we cultured small clusters of MZ*fz7a/b* mutant ppl progenitor cells expressing Opto-Fz7 on Fibronectin-coated substrates and determined how local activation

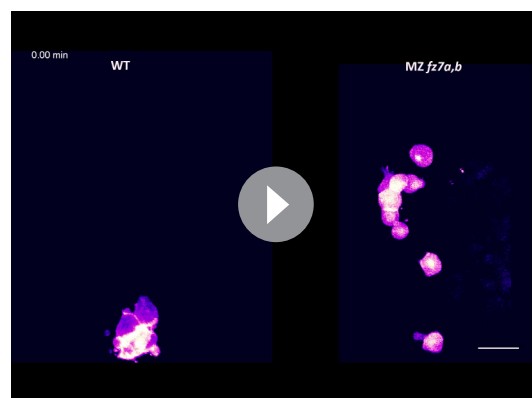

**Video 3.** MZ*fz7a/b* mutant and wt donor ppl cells (magenta) transplanted into the ppl of a Tg(*gsc:GFP-CAAX*) host embryo with the same genotype (homotypic transplantations). Donor cells are marked by mkate2-CAAX expression. Only donor cells are shown. Yellow arrowheads point to filopodia, white arrowheads to larger lamellipodia/pseudopodia-like protrusions. Scale bar, 30 µm.
DOI: https://doi.org/10.7554/eLife.42093.011

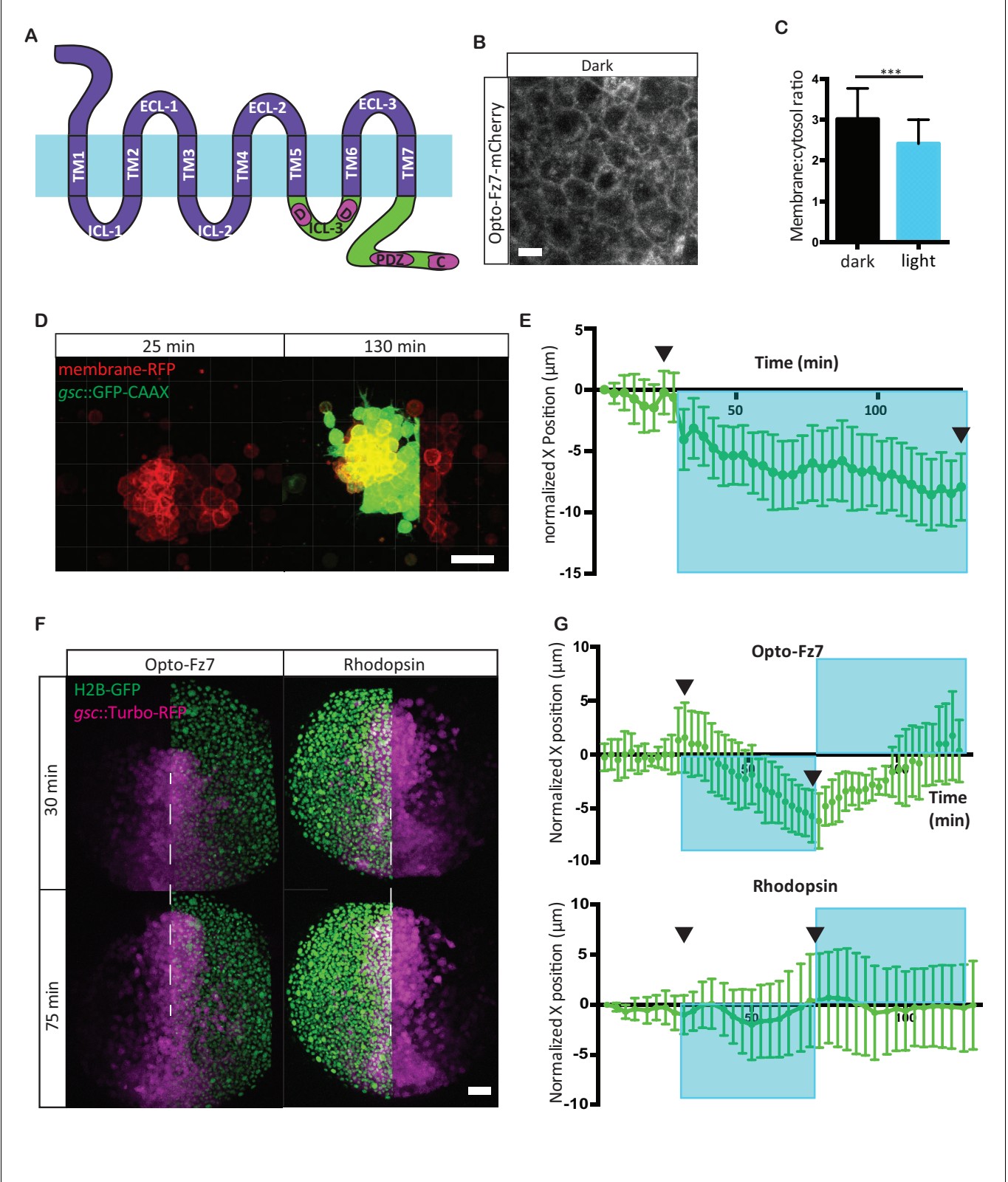

**Figure 4.** Photo-activation of Fz7 signaling in prechordal plate tissues in vitro and in vivo using Opto-Fz7. (**A**) Schematic of the Opto-Fz7 receptor. Light blue bar symbolizes the plasma membrane with extracellular space above and intracellular space below the bar. Rhodopsin parts are shown in purple and Fz7 parts in green. TM1-7, transmembrane domains 1–7; ECL 1–3, extracellular loops 1–3, ICL 1–3, intracellular loops 1–3. Fz7 domains contain two Dshbinding sites in ICL3 (**D1, D2**), a PDZ domain-binding site (PDZ) near the C-terminus, and the last four amino acids (ETTV), which also have Dsh-
*Figure 4 continued on next page*

*Figure 4 continued*

binding function. (B) Confocal image of Opto-Fz7 subcellular localization in embryos injected with *opto-Fz7-mcherry* mRNA and 9-cis-retinal at 70% epiboly kept in dark. Scale bar, 20 µm. (C) Cytosol to membrane fluorescence intensity ratios of Opto-Fz7 shown in (B). Error bars are standard deviations; N = 67 cells from 10 embryos in the dark group, and 108 cells from 12 embryos in the light group, ***p<0.001 (Student's t-test). (D) Explants of ppl tissue from MZ*fz7a/b*;Tg(*gsc:GFP-CAAX*) mutant embryos expressing membrane-RFP, Opto-Fz7 and injected with 9-cis-retinal. Opto-Fz7 is photoactivated using a 488 nm laser in the left half of the explant (shown by GFP signal). Left panel, directly before activation; right panel, 130 min after activation. Note that the explant has moved into the activated region. Scale bar, 50 µm. (E) Movement of the explant along the x-axis. The x-position at the start of the experiment is used as x = 0 position. Blue box marks the time-interval of Opto-Fz7 photoactivation in (D). Arrows mark the time points when the images in (D) were taken. Eerror bars are standard error of mean; N = 15 explants from 2 experiments. (F) Confocal images of the ppl in MZ*fz7a/b*;Tg(*gsc:TurboRFP*) embryos at 80% epiboly stage with animal to the top. Only one half of the ppl is exposed to 488 nm light to activate Opto-Fz7 signaling. Embryos were injected with 9-cis-retinal, and all nuclei in the right half of the ppl embryo are marked by H2B-GFP expression (to outline the initial photoactivation domain). Experimental embryos express Opto-Fz7 (left column) and control embryos Rhodopsin (right column). Upper panels are at the start of photoactivation (30 min after the experiment started), lower panels after 45 min of photoactivation (75 min after the experiment started). Dashed lines mark the ppl midline at the start of photactivation. Scale bar, 50 µm. (G) Ppl movement along the x-axis of the ppl. The x-position atthe time of photoactivation is used as x = 0 position. Error bars are standard deviations. Blue boxes outline time intervals of photoactivation in (F). Arrows mark the time points when the images in (F) were taken. N = 6 embryos per condition.

DOI: https://doi.org/10.7554/eLife.42093.012

The following source data and figure supplements are available for figure 4:

**Source data 1.** Opto-Fz7 characterization and its effect on cell migration upon spatially restricted activation.
DOI: https://doi.org/10.7554/eLife.42093.018
**Figure supplement 1.** Opto-Fz7 architecture.
DOI: https://doi.org/10.7554/eLife.42093.013
**Figure supplement 2.** Receptor structural analysis.
DOI: https://doi.org/10.7554/eLife.42093.014
**Figure supplement 3.** Opto-Fz7 subcellular localization.
DOI: https://doi.org/10.7554/eLife.42093.015
**Figure supplement 4.** Effect of Opto-Fz7 activation on non-canonical Wnt/PCP and canonical Wnt signaling.
DOI: https://doi.org/10.7554/eLife.42093.016
**Figure supplement 5.** Effect of Opto-Fz7 activation on ppl progenitor protrusive activity.
DOI: https://doi.org/10.7554/eLife.42093.017

of Fz7 by light in approximately half of the cluster would affect cluster protrusion formation and movement. We found that local activation of Opto-Fz7 in these clusters triggered movement of the entire cell cluster in the direction of the activated region within the cluster (*Figure 4D&E*, *Video 4*, *Figure 4—source data 1*). This cluster movement was accompanied by increased protrusion formation at the light-exposed side of the explant (*Figure 4—figure supplement 5A&B*, *Figure 4* – source data, *Video 4*), suggesting that localized stimulation of cell motility by unilateral Opto-Fz7 activation triggers this movement. To test how this in vitro observation relates to the situation in vivo, we expressed Opto-Fz7 in MZ*fz7a/b* mutant embryos and activated Fz7 in one half of the ppl. We then asked whether Fz7 activation of half of the ppl would redirect ppl cell movements. Strikingly, we found that the ppl motion was biased to the direction of the activated side (*Figure 4F&G*, *Figure 4—figure supplement 5E&F*, *Figure 4* – source data, *Video 5*), an effect that could be effectively reverted by switching the activation of Fz7 to the initially non-activated side of the ppl (*Figure 4G*, *Figure 4* – source data, *Video 5*). Furthermore, comparing protrusion formation at the leading edge of the activated versus non-

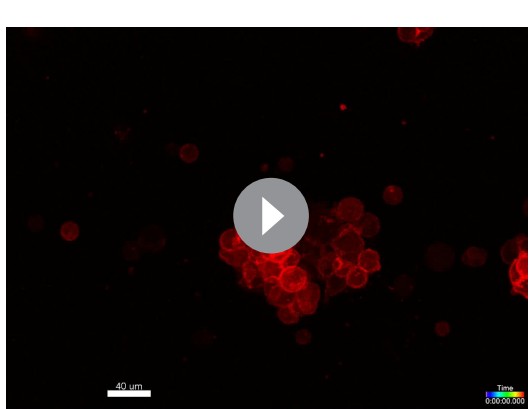

**Video 4.** Cultured ppl explant from MZ*fz7a/b*;Tg(*gsc: GFP-CAAX*) mutant embryos expressing membrane-RFP and Opto-Fz7 and injected with 9-cis-retinal. Upon photoactivation of Opto-Fz7 in half of the explant (shown by GFP excitation), the explant begins to move into the light-exposed area. Scale bar, 30 µm.
DOI: https://doi.org/10.7554/eLife.42093.019

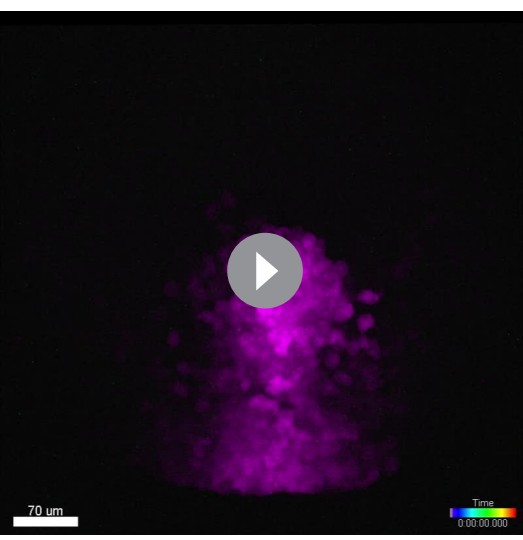

**Video 5.** Ppl movement in a MZ*fz7a/b*; Tg(*gsc:Turbo-RFP*) embryo expressing H2B-GFP (marking all nuclei, green) and Opto-Fz7 and injected with 9-cis-retinal. Ppl cells are marked by expression of *gsc*:Turbo-RFP (magenta). Upon photoactivation of Opto-Fz7 in half of the ppl, the explant begins to move into the light-exposed area. Nuclei are only shown in the light-activated side of the ppl to outline the activation domain. Scale bar, 70 µm.

DOI: https://doi.org/10.7554/eLife.42093.020

activated side of the ppl showed a higher number of lamellipodia-like protrusions at the activated side (*Figure 4—figure supplement 5C&D*, *Figure 4—source data 1*), suggesting that Fz7 redirects ppl migration by triggering protrusion formation locally within the ppl. In contrast, control MZ*fz7a/b* mutant embryos expressing Rhodopsin instead of Opto-Fz7 did not show any redirection of ppl cell migration upon half-sided light stimulation (*Figure 4F&G*, *Figure 4—figure supplement 5E&F*, *Figure 4—source data 1*). Collectively, these findings suggest that locally restricted activation of Fz7 has the capability of redirecting ppl cell migration.

Our data so far demonstrate that exogenous Fz7 activation can control the migration direction of ppl progenitor cells, pointing to the possibility that localized endogenous Fz7 activation may direct ppl progenitor cell protrusion formation and migration. To address this possibility, we sought to uniformly activate Fz7signaling in ppl progenitors both in vitro and in vivo and determine whether such uniform activation is compatible with ppl cell protrusion formation and directed migration. We reasoned that in case localized endogenous Fz7 activation directs ppl progenitor cell protrusion formation and migration, then uniform activation in MZ*fz7a/b* mutants would be insufficient to restore these processes. First, we asked whether reduced protrusion formation at the margin of small clusters of MZ*fz7a/b* mutant ppl progenitors in culture could be restored by uniform activation of Fz7 signaling. Protrusion formation was fully rescued in cultured clusters of MZ*fz7a/b* mutant ppl progenitors upon uniform light activation of Opto-Fz7 in these cells (*Figure 5A&B*, *Figure 5—source data 1*). Moreover, uniform over-activation of Fz7signaling in wt ppl cell clusters by uniform light activation of Opto-Fz7 led to reduced protrusion formation (*Figure 5A&B*, *Figure 5—source data 1*), consistent with previous observations that over-activation of Wnt-Fz/PCP signaling mimics the loss-of-function phenotype (*Heisenberg et al., 2000*; *Kilian et al., 2003*). This points to the possibility that Fz7 signaling has a mere permissive function in ppl cell protrusion formation. To further test this possibility, we also attempted to rescue the ppl cell movement phenotype in MZ*fz7a/b* mutant embryos by uniform activation of Opto-Fz7. Strikingly, we found that uniform activation of Fz7 signaling in MZ*fz7a/b* mutant embryos expressing Opto-Fz7 fully rescued ppl movement as judged by ppl shape and distance to the anterior tip of the notochord (*Figure 5C–E*, *Figure 5—source data 1*). This clearly argues against a critical requirement of localized Fz7 activation for directed ppl cell migration. Likewise, we found that uniform activation of Fz7 in ppl cells could fully rescue their cell-autonomous migration and protrusion formation defects when transplanted into the ppl of a wt embryo (*Figure 5F–I*, *Figure 5—figure supplement 1*, *Video 6 and 7*, *Figure 5—source data 1*). In contrast, no such rescue activity was detected in control conditions (absence of light, Rhodopsin overexpression together with 9-cis-retinal injection, and 9-cis-retinal injection alone; *Figure 5C–E*, *Figure 5—figure supplement 2*, *Figure 5—source data 1*). Together, these findings suggest that Fz7 signaling has a permissive function in ppl cell protrusion formation and directed migration during gastrulation.

## Discussion

The mechanisms by which non-canonical Wnt-Fz/PCP signaling affects mesenchymal cell polarization, directed migration, and intercalation are still debated. There is increasing evidence that non-

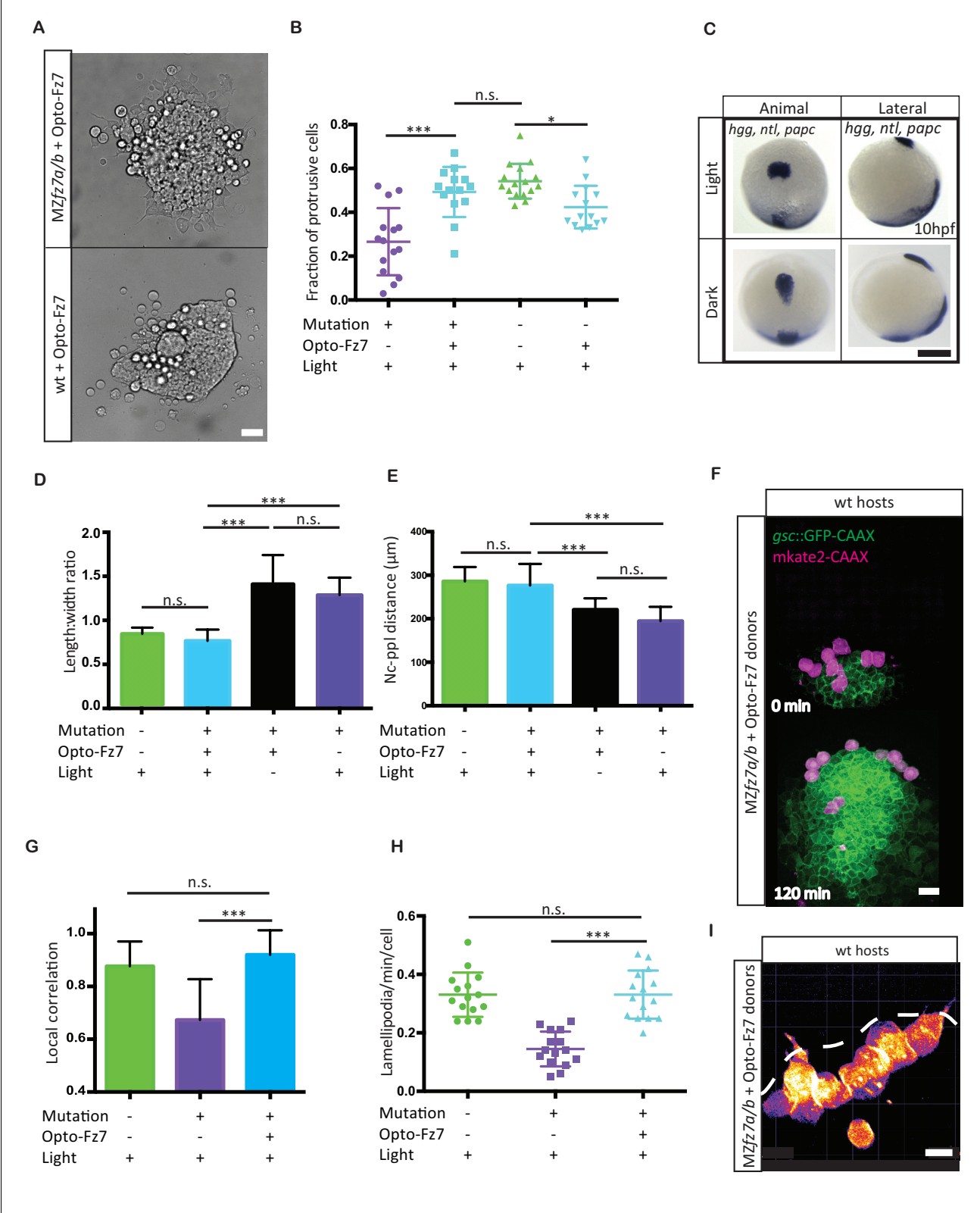

**Figure 5.** Uniform activation of Opto-Fz7 signaling in MZ*fz7a/b* mutants. (**A**) Bright-field images of ppl explants from wt and MZ*fz7a/b* mutant embryos both expressing Opto-Fz7 and injected with 9-cis-retinal. Scale bar, 40 μm. (**B**) Fraction of ppl cells at the edge of the explants shown in (**A**) displaying lamellipodia-like protrusions after 4 hrs in culture. Opto-Fz7 expressing explants were compared to MZ*fz7a/b* and wt explants shown in *Figure 3B*. N = 12 explants per genotype, ***p<0.001, *p<0.05, and n.s., non significant (ANOVA followed by Tukey's multiple comparison test). (**C**)
*Figure 5 continued on next page*

*Figure 5 continued*

Animal (left column) and lateral (right column) views of the notochord (nc) and prechordal plate (ppl) in MZ*fz7a/b* mutant embryos expressing Opto-Fz7 and injected with 9-cis-retinal at the end of gastrulation (bud stage, 10hpf) labeled by in situ hybridization for *hgg* (ppl), *ntl* (nc), *and papc* (paraxial mesoderm). Embryos were either exposed to light (upper row) or kept in the dark (lower row). Scale bar, 250 μm. (D,E) Length-to-width ratio of the ppl (D) and distance between the anterior end of the nc and the posterior end of the ppl (E) in MZ*fz7a/b* mutant embryos expressing Opto-Fz7 and injected with 9-cis-retinal either exposed to light (blue) or kept in the dark (black). Wt (green) and untreated MZ*fz7a/b* mutant (purple) embryos were included as controls. Error bars are standard deviations. N = 25 (wt), 24 (MZ*fz7a/b*), 36 (Opto-Fz7, light), 36 (Opto-Fz7, dark), ***p<0.001, and n.s., non significant (Kruskal-Wallis test followed by Dunn's multiple comparison test). (F) Confocal images of the ppl in wt host embryos containing transplanted MZ*fz7a/b* mutant donor cells (magenta) expressing Opto-Fz7 and mkate2-CAAX and injected with 9-cis-retinal. Dorsal views with animal pole to the top. Host cells are marked by *gsc*:GFP-CAAX expression (green). Scale bar, 40 μm. (G) Local correlation between host and donor ppl cell movements shown in (F), compared to MZ*fz7a/b* and wt ppl donor cells shown in *Figure 2C*. Error bars are standard deviations. N = 22 (wt), 36 (mutants), and 35 (mutants expressing Opto-Fz7) binned sectors from 3 embryos each, ***p<0.001, and n.s., non significant (Kruskal-Wallis test followed by Dunn's multiple comparison test). (H) Number of lamellipodia-like protrusions per cell and minute in the transplanted donor cells shown in (I), compared to MZ*fz7a/b* and wt ppl donor cells shown in *Figure 3D*. N = 15 cells per genotype, ***p<0.001, and n.s., non significant (ANOVA followed by Tukey's multiple comparison test). (I) Confocal images of transplanted MZ*fz7a/b* mutant donor ppl cells expressing OptoFz7 and mkate2-CAAX and injected with 9-cis-retinal within the ppl of a wt host embryo (anterior edge outlined by dashed white line). Scale bar, 20 μm.

DOI: https://doi.org/10.7554/eLife.42093.021

The following source data and figure supplements are available for figure 5:

**Source data 1.** Rescue of MZ*fz7a/b* embryos by Opto-Fz7 activation.
DOI: https://doi.org/10.7554/eLife.42093.026

**Figure supplement 1.** Protrusion formation in MZ*fz7a/b* cells after Opto-Fz7 activation.
DOI: https://doi.org/10.7554/eLife.42093.022

**Figure supplement 2.** Opto-Fz7 control experiments.
DOI: https://doi.org/10.7554/eLife.42093.023

**Figure supplement 3.** Convergence and extension movements in MZ*fz7a/b* mutants after Opto-Fz7 activation.
DOI: https://doi.org/10.7554/eLife.42093.024

**Figure supplement 4.** Subcellular localization of PCP components in the ppl.
DOI: https://doi.org/10.7554/eLife.42093.025

canonical Wnt-ligands, including Wnt11, are required for directed cell migration, and that gradients of exogenous Wnt11 can direct elongation of myogenic cells. However, whether and how endogenous Wnt11 directs cell migration remains unclear.

To control Wnt-Fz/PCP signaling in a ligand-independent manner, we developed Opto-Fz7, a light-activated variant of the Wnt receptor Fz7, which permits spatio-temporally controlled and ligand-independent Wnt-Fz/PCP signaling activation. The advantage of Opto-Fz7 over, for example, constitutive active versions of this receptor is not only that Opto-Fz7 allows spatio-temporally restricted activation, but also that Opto-Fz7 is entirely insensitive to activation by endogenous Wnt ligands within the embryo. This excludes any potential polarizing influence of endogenous Wnt ligands on Fz7 signaling activity and thus permits truly uniform signaling activation to distinguish between instructive versus permissive functions of Fz7 signaling within the embryo. Our results highlight how this light-activated receptor in the context of the transparent zebrafish model system can be used to determine how signals shape cellular processes during development (*Sako et al., 2016*; *Johnson et al., 2017*; *Izquierdo et al., 2018*).

Our findings that Wnt11 and Fz7 are required for directed ppl cell migration, and that uniform activation of Fz7 signaling in ppl cells is sufficient for their directed migration, suggest that any

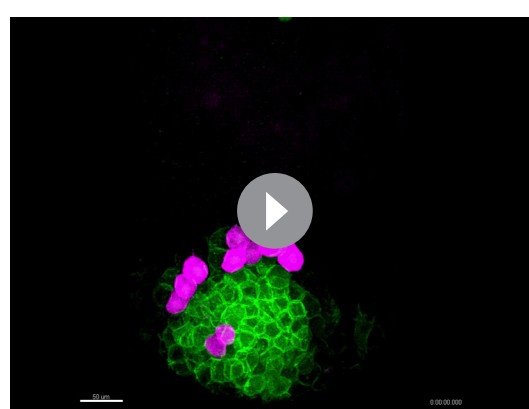

**Video 6.** MZ*fz7a/b* donor cells expressing Opto-Fz7 and injected with 9-cis-retinal transplanted into the ppl of a wt Tg(*gsc:GFP-CAAX*) host ppl. Transplanted cells are marked by mkate2-CAAX expression (magenta) and donor cells express *gsc*:GFP-CAAX (green). Scale bar, 50 μm.
DOI: https://doi.org/10.7554/eLife.42093.027

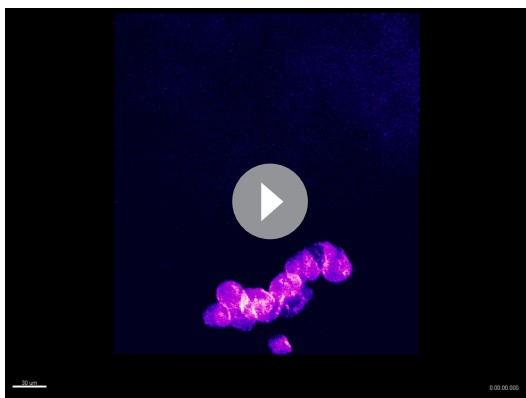

**Video 7.** Movement of MZ*fz7a/b* mutant donor ppl cells expressing Opto-Fz7, injected with 9-cis-retinal and transplanted into the ppl of a wt Tg(*gsc:GFP-CAAX*) host. Transplanted cells are marked by mkate2-CAAX expression (magenta) and donor cells express *gsc*:GFP-CAAX (green). Yellow arrowheads point to filopodia, white arrowheads to larger lamelipodia/pseudopodia-like protrusions. Scale bar, 30 µm.
DOI: https://doi.org/10.7554/eLife.42093.028

potentially instructive function of Fz7 signaling in directing ppl cell migration is dispensable. These results are consistent with recent suggestions that polarization of chondrocytes within the developing mouse limb bud is achieved by the combination of permissive Wnt5 and instructive FGF signals (*Gao et al., 2018*), although direct evidence for the purely permissive nature of Wnt5 signaling is still missing.

While our findings clearly show that Fz7-mediated Wnt-Fz/PCP signaling is required permissively for ppl cell migration, they do not exclude the possibility of any other receptor activating Wnt-Fz/PCP signaling in ppl cells, which as such, might still have an instructive function in directing ppl cell migration. For instance, the receptor-tyrosine kinases Ror2, Ryk and Ptk7 have previously been shown to functionally interact with Wnt11 and/or Fz7 in regulating non-canonical Wnt signaling (*Kim et al., 2008b*; *Gao et al., 2011*; *Yen et al., 2009*; *Hayes et al., 2013*; *Brinkmann et al., 2016*), pointing to the possibility that they might also be involved in mediating Wnt11 signaling within the ppl. Also, uniform activation of Fz7 could not rescue the convergence phenotypes of the notochord and the paraxial mesoderm (*Figure 5—figure supplement 3*, *Figure 5—source data 1*), suggesting that the mode of Fz7 function might be context-dependent.

While our data exclude a direct function of localized Fz7 signaling in polarizing ppl cell migration, Fz7 signaling in ppl cells is still essential for this process. Our finding that Fz7 signaling is required for the formation of actin-rich lamellipodia-like protrusions in ppl cells, suggests that Fz7 enables directed ppl cell migration by promoting protrusion formation. How Fz7 regulates this process is yet unclear, but previous findings, suggesting that non-canonical Wnt-Fz/PCP signaling can activate small Rho GTPases known to control protrusion formation during migration (*Marlow et al., 2002*; *Eaton et al., 1996*), point to Rho GTPases as potential downstream effectors of Fz7 in this process.

A core mechanism by which Wnt-Fz/PCP signaling polarizes epithelial cells is the polarized distribution of different Wnt-Fz/PCP components along the polarization axis (*Collu and Mlodzik, 2015*). While this is well documented in epithelial cells, it is less clear whether there is any polarized subcellular distribution of Wnt-Fz/PCP components in mesenchymal cells (*Roszko et al., 2015*; *Jussila and Ciruna, 2017*). Studies in zebrafish suggest that in neurectoderm progenitor cells, the core Wnt-Fz/PCP component Pk accumulates at the anterior side, while the Wnt-Fz/PCP signaling mediator Dsh localizes to the posterior side of these cells (*Ciruna et al., 2006*; *Yin et al., 2008*). In contrast, we found no evidence for a clearly recognizable polarized subcellular distribution of Fz7 or any other Wnt-Fz/PCP components in ppl cells (*Figure 4B*, *Figure 5—figure supplement 4*, *Figure 5—source data 1*), suggesting that endogenous Wnt-Fz/PCP signaling might be uniform rather than polarized in these cells. Yet, local variations in PCP component turnover could contribute to dynamic changes in polarized distribution of PCP components (*Strutt et al., 2011*; *Chien et al., 2015*; *Shi et al., 2016*) that may not have been detected here.

Notably, while our data suggest that Wnt11/Fz7 signaling is required for directed ppl cell migration, ppl cells do not endogenously express Wnt11. However, Wnt11 is expressed in the anterior paraxial mesendoderm close to the ppl (*Heisenberg et al., 2000*; *Kilian et al., 2003*), pointing to the possibility that Wnt11 secreted by the anterior paraxial mesendoderm acts on adjacent ppl cells, thereby promoting protrusion formation and directed migration of these cells. Moreover, it is conceivable that Fz7 is not exclusively functioning as a receptor for Wnt11, and that other non-canonical Wnt ligands expressed within the ppl activate Fz7 signaling there.

Non-canonical Wnt-Fz/PCP signaling has long been thought to function as a prime signal polarizing cells in development (*Yang and Mlodzik, 2015*; *Collu and Mlodzik, 2015*). While such function

has been well established in epithelial cells (*Dabdoub et al., 2003*; *Wu et al., 2013*; *Chu and Sokol, 2016*), the role of non-canonical Wnt-Fz/PCP signaling in mesenchymal cells is less understood. Our findings show that signaling through the Wnt receptor Fz7, although being required for mesenchymal cell polarization and directed migration, has no direct function in polarizing those cells; it rather promotes protrusion formation, allowing the cells to undergo directed migration. This suggests that non-canonical Wnt-Fz/PCP signaling can have very different functions depending on the specific cellular context in which it acts.

# Materials and methods

## Key resources table

| Reagent type (species) or resource | Designation | Source or reference | Identifiers | Additional information |
|---|---|---|---|---|
| Experimental model | Zebrafish: *Tg(gsc::EGFP-CAAX)* | *Smutny et al., 2017* | NA | |
| Experimental model | Zebrafish: *Tg(gsc::Turbo-RFP)* | *Sako et al. (2016)* | ZDB-ALT-161005–1 | |
| Experimental model | Zebrafish: MZfz7a/b | *Quesada-Hernández et al. (2010)* | ZDB-FISH-150901–19586 | |
| Experimental model | Zebrafish: MZfz7a/b;Tg(gsc::EGFP-CAAX) | this paper | NA | |
| Experimental model | Zebrafish: MZfz7a/b;Tg(gsc::Turbo-RFP) | this paper | NA | |
| Oligonucleotides | *casanova morpholino GCATCCGGTCG AGATACATGCTGTT* | Gene Tools | ZDB-MRPHLNO-050818–2 | |
| Recombinant DNA reagent | pcDNA 3.1-Opto-Fz7 plasmid for mRNA synthesis | this paper | NA | |
| Recombinant DNA reagent | pcDNA 3.1-Opto-Fz7-mCherry plasmid for mRNA synthesis | This paper | NA | |
| Recombinant DNA reagent | pCS2-Ndr2 plasmid for mRNA synthesis | *Rebagliati et al., 1998* | NM_139133.1 | |
| Recombinant DNA reagent | pCS2-Wnt11-3xmCherry | this paper | NM_001144804.1 | |
| Recombinant DNA reagent | pCS2-Fz7a-mNeongreen | this paper | NM_131139.2 | |
| Recombinant DNA reagent | pCS2-Pk1-venus | *Carreira-Barbosa et al., 2003* | NM_183342.2 | |
| Recombinant DNA reagent | pCS2-Dvl2-eGFP | other | NM_212648.1 | courtesy of Masazumi Tada |
| Antibody | ß-catenin clone 15B6 | Sigma | C7207, RRID:AB_476865 | |
| Antibody | Alexa Fluor 568 Goat Anti-Mouse IgG (H + L) Antibody | Molecular Probes | A11004, RRID: AB_141371 | |
| Antibody | GFP mAb3E6 | Molecular Probes | A11120, RRID:AB_221568 | |
| Antibody | Alexa Fluor 488 Goat Anti-Mouse IgG (H + L) Antibody | Molecular Probes | A11001 | |

*Continued on next page*

*Continued*

| Reagent type (species) or resource | Designation | Source or reference | Identifiers | Additional information |
|---|---|---|---|---|
| chemical compound | 9-cis-retinal | Sigma | R5754 | |
| Cell culture reagent | DMEM/F12 | Sigma | DF-041-B | |

## Opto-Fz7 construct

Opto-Fz7 was designed based on zebrafish Fz7 (NCBI NM_131139.1) and bovine Rhodopsin (NCBI NM_001014890.2). The transmembrane domains of Rhodopsin and the intracellular domains of Fz7 were predicted based on published data of chimeric proteins (*Liu et al., 2001*; *Kim et al., 2005*; *Airan et al., 2009*). ICL3 was modified according to a mutational study of Fz proteins (*Tauriello et al., 2012*). Cloning was performed using the golden gate method (*Engler and Marillonnet, 2014*; *Engler et al., 2009*).

## Structural analysis of GPCRs

The coordinates for the crystal structures of bovine Rhodopsin and human and *Xenopus* Smo were obtained from the Protein Data Bank (PDB; http://www.rcsb.org). The structures corresponded to either inactive (Rho: 1U19, Smo: 4JKV) or active (Rhodopsin: 3PQR, Smo: 6D32) states (*Okada et al., 2004*; *Choe et al., 2011*; *Wang et al., 2013*; *Huang et al., 2018*). Visualization of protein structures, structural superimpositions, and structure-based sequence alignments were performed using UCSF Chimera (*Pettersen et al., 2004*). Residue contacts within each structure were obtained using the Protein Contacts Atlas (*Kayikci et al., 2018*). Only contacts mediated by residues between TM1-7 or helix eight according to GPCRdb (*Isberg et al., 2015*) numbering scheme were considered. Contact distribution was calculated as the number of contacts for each helix pair compared to the total number of contacts found in each receptor. Contact distributions were compared between GPCRs of the same activation state.

## Fish lines and husbandry

Zebrafish handling was carried out as described (*Westerfield, 2000*). Embryos were raised at 28–31° and staged according to *Kimmel et al. (1995)*. The following fish lines were used: Wild type ABxTL hybrids, Tg(*gsc::GFP-CAAX*) (*Smutny et al., 2017*), Tg(*gsc::TurboRFP*) (*Sako et al., 2016*; *Barone et al., 2017*), MZ*fz7a/b -/-* (*Quesada-Hernández et al., 2010*), MZ*fz7a/b -/-*; Tg(*gsc::GFP-CAAX*), and MZ*fz7a/b -/-*; Tg(*gsc::TurboRFP*). Mutant transgenic lines were generated by crossing transgenic lines to the MZ*fz7a/b -/-* line.

## Whole mount in situ hybridization (WMISH)

WMISHs were performed as described (*Smutny et al., 2017*). RNA antisense probes were synthesized from partial sequences of cDNA using mMessage mMachine kits (ThermoFisher AM1344) with a Roche Dig-labeling mix (Sigma 11277073910). Images were taken using a stereo-microscope (Olympus SZX 12) equipped with a QImaging Micropublisher 5.0 camera.

## Whole mount antibody staining

Antibody stainings were performed as previously described (*Compagnon et al., 2014*), except that an extra step of permeabilization by 100% MeOH (60 min at −20°C) was introduced. Anti β-catenin (Sigma C7207) antibody was used at 1:250 dilution, anti GFP (Molecular Probes A11120) at 1:50.

## Injection of mRNA and morpholino antisense oligonucelotides

Embryos consisting of ppl progenitors only were obtained by injecting 100 pg *ndr2* (*cyclops*) mRNA and 2 ng *sox32* (*casanova*) morpholino into 1-cell stage wt or MZ*fz7a/b* embryos as described (*Krieg et al., 2008*). For visualization of plasma membrane and nuclei, 100 pg *mkate2-CAAX* mRNA and 50 pg of *H2A-mCherry* or *H2B-BFP* mRNA, respectively, were injected at the 1-cell stage. For the protrusion quantification experiments, the plasma membrane was labeled by injection of 250 pg

of *mkate2-CAAX* mRNA at the 1-cell stage. For optogenetic activation of the Wnt/PCP pathway, 20 pg of o*pto-Ffz7* mRNA was injected. For analyzing Opto-Fz7 subcellular localization, 200 pg *Opto-Fz7-mcherry* mRNA was injected at the 1-cell stage. *Opto-fz7* mRNA injections were generally combined with injection of 140 pg of 9-cis-retinal. For determining the subcellular localization of Wnt-Fz/PCP pathway components, embryos were injected with 60 pg *wnt11-3x-mCherry*, 25 pg *pk1-venus*, 100 pg *fz7a-mNeongreen*, or 100 pg *dvl2-GFP* mRNA. mRNAs were synthesized using SP6 mMessage mMachine kit (ThermoFisher AM1340) from linearized DNA templates.

## Ppl explant cultures

The in vitro cell and explant assays were performed as described (*Ruprecht et al., 2015*; *Krens et al., 2017*). Embryos consisting of only ppl progenitors were induced as described above. Glass bottom dishes (CellView Cell Culture Dishes, Glass Bottom, four compartment, Greiner 627975) were plasma-cleaned (2 min, maximum level) and coated with Fibronectin (Sigma F1141). For coating, 50 µl of a diluted Fibronectin stock (1:4 in distilled water) was applied to the dish and allowed to dry. When uninjected control embryos were at dome stage, the ppl-induced embryos were transferred to glass dishes containing Danieus medium (58 mM, 0.7 mM KCl, 0.4 mM MgSO$_4$, 0.6 mM Ca(NO$_3$)$_2$, 5 mM HEPES, pH 7.6). After dechorionation, ppl explants were prepared from ppl-induced embryos by removing the blastoderm from the yolk, transferring it to glass dishes with pre-warmed DMEM cell culture medium (stock diluted with 10% distilled water, Sigma D6434) and cutting it into smaller pieces consisting of ≈ 50 cells. The dissected explants were then transferred to the Fibronectin-coated glass bottom dishes, incubated in culture medium at 28°C for 2 hrs and subsequently imaged using a Visitron TIRF/FRAP epifluorescence microscope with bright-field and 475 nm illumination.

## Cell transplantations

For transplantation experiments, donor embryos consisting of ppl cells only were induced as described above. As host embryos, MZ*fz7a/b;* Tg(*gsc:GFP-CAAX*) or Tg(*gsc:GFP-CAAX*) embryos were used. At 5 hpf, host and donor embryos were dechorionated in E3 buffer (5 mM NaCl, 0.17 mM KCl, 0.33 mM CaCl$_2$ × 2 H2O, 0.33 mM MgSO$_4$ × 7 H2O, pH 6.5) in glass dishes. At 6 hpf, donor and host embryos were aligned in small grooves in agarose-coated dish containing E3 buffer. For transplantations, groups of 5–20 cells were typically aspirated from the donor embryos using a transplantation needle (20 µm diameter, with spike, Biomedical instruments) and then transplanted to the leading edge of the ppl of host embryos.

## Confocal and 2-photon imaging

Embryos were dechorionated and then mounted in 0.5% low melting point agarose in E3 buffer within 2% agarose molds inside a petri dish. Imaging of live samples was performed using a LaVision Trim 2-photon microscope equipped with a Zeiss Plan-Apochromat 20x/1.0 water immersion objective and Ti:Sa laser (Chameleon, Coherent) set at 810 nm and OPO laser set at 1150 nm. Fixed samples were mounted in agarose as described for live samples (except that PBS was used as buffer) and imaged using a Zeiss LSM 880 or 800 upright microscope equipped with a Zeiss Plan-Apochromat 20x/1.0 water immersion objective.

## Optogenetic procedures

For embryo rescue experiments, embryos were injected with 20 pg of *opto-fz7* mRNA at the 1 cell stage. The injected embryo batch was split into two groups of equal size and kept in E3 buffer. The control batch was put into a plastic box and wrapped in aluminum foil to block light exposure. Both batches were incubated together from the 2-cell stage onwards at 28 or 31°C in an incubator equipped with 500 nm light emitting diodes (LEDs) with an intensity of 3 µW/mm$^2$. At bud stage (10 hpf), the samples were fixed in 4% PFA. For analyzing Opto-Fz7 subcellular localization, embryos were processed as described above and then fixed at 70% epiboly. For localized Opto-Fz7-activation, injected embryos were protected from light and incubated until the desired stage. Subsequent embryo mounting and imaging were performed under red light illumination to avoid premature activation of Opto-Fz7. Local Opto-Fz7 activation experiments were performed on a Leica SP5 upright microscope using the Live Data Mode plugin and a 25x/0.95 NA water-dipping lens at a

radiant exposure of 0.013nj/$\mu$m$^2$. General imaging of these experiments was performed using a 561 nm laser to avoid Opto-Fz7 activation (*Xu et al., 2014*), and local activation of Opto-Fz7 was performed with a 488 nm laser. For general imaging and local Opto-Fz7 activation in cultured explants, an inverted Leica SP5 microscope equipped with a 20x/0.7 NA air objective at a radiant exposure of 0.009 nj/$\mu$m$^2$ was used. For ubiquitous activation of Opto-Fz7 in vitro, explants were prepared and incubated as described above and then imaged using a Visitron TIRF/FRAP epifluorescence microscope with brightfield and 475 nm illumination at a radiant exposure of 0.018nj/$\mu$m$^2$. For imaging of the ubiquitous activation in vivo experiments, a LaVision Trim 2-photon microscope with 800 nm, 830 nm and 1100 nm illumination at a radiant exposure of 0.044 nj/$\mu$m$^2$ was used.

## Statistics

Statistical analysis was done using GraphPad Prism 6. Rose diagrams were done using R. Statistical details of experiments are reported in the figures and figure legends. Sample size is reported in the figure legends and no statistical test was used to determine sample size. The biological replicate is defined as the number of embryos or independent batches of embryos, as stated in the figure legends. No inclusion/exclusion or randomization criteria were used and all analysed samples are included. To test for normality of a sample, a D'Agostino and Pearson omnibus normality test was used. In case two samples were compared and normal distribution was assumed, an unpaired t-test was performed. If two samples were not normally distributed, a Mann-Whitney test was performed, instead. In case more than two normally distributed samples were compared, an ANOVA was performed followed by Tukey's multiple comparison test. If no normal distribution could be assumed, a Kruskal-Wallis test followed by Dunn's multiple comparison test was used. No blind allocations were used during the experiments or in the analysis

## Image processing

Images acquired by multiphoton or confocal microscopy were imported to Imaris (Bitplane) to 3D-visualize the recorded channels for further analysis. For correlation analysis, Imaris 9 was used to orient all embryos along the AV axis at the dorsal side, which was identified by the *gsc*:GFP-CAAX signal. Ppl nuclei were separated from other nuclei by surface masking of the *gsc*:GFP channel, and neuroectoderm nuclei were identified by negative surface masking and position relative to the ppl. Nuclei positions of ppl progenitors and neuroectoderm cells in xyz-dimensions were exported for each time point and used for the correlation analysis. For the analysis of protrusive behavior in vivo, images were processed using Imaris 9, and the reconstructed movies were exported as tiff-stacks and further analyzed in Fiji. $\beta$-catenin data were analyzed by masking the nuclear signal based on an additional DAPI channel and measuring the mean intensity of the channel.

## Correlation analysis

Local movement correlation between ppl and neurectoderm cells was determined as described (*Smutny et al., 2017*). In brief, 3D velocity vectors were averaged in sectors of 50 by 50 $\mu$m in xy planes and the full z direction for every time point. The sectors were arranged in a grid that covered the whole imaged area and was centered on the leading edge of the ppl. To quantify the correlation between the ppl and the overlying ectoderm, the directional correlation of the two was calculated for every sector of the grid. The correlation could adopt values from 1 (ppl and the neuroectoderm move in the same direction) to $-1$ (ppl and neurectoderm move in opposite directions).

## Analysis of protrusion frequency and orientation

In the in vivo experiments, the number of each type of protrusion per cell was counted in movies of 1–2 hrs and normalized to the length of the movie, yielding the metric of protrusions per cell and minute. The orientation of protrusions was measured using the Fiji angle tool with the animal pole direction set to 0°. The fraction of marginal cells displaying protrusions in cell culture experiments was determined by counting the number of cells with or without protrusions and plotting the ratio of these numbers.

## Fiji analysis of in situ hybridized embryos, cell culture experiments, and subcellular localization of fluorecently tagged proteins

To quantify the expression domains of marker genes in in situ hybridized embryos in a semi-automated fashion, we used two custom-made Fiji scripts (Gabriel Krens, IST Austria Imaging Facility, source code files ppl2notochord and ppl_LW-ratio). For measuring the distance between notochord and prechordal plate, a region of interest (ROI) was selected in lateral images that contained the posterior end of the ppl (marked by *hgg* expression) and the anterior end of the notochord (marked by *ntl* expression), with the pixel size normalized to a scale bar that was added during image acquisition. The script then automatically measured the distance between the two expression domains. For measuring the length to width ratio (LWR) of the ppl, the script automatically measured the length of longest axis of the ppl (marked by *hgg* expression) and of the axis perpendicular to the longest axis, and then calculated the ratio of the two axis lengths. The membrane to cytosol ratio in the protein localization experiments was done by drawing a line over several cells, measuring the fluorescence intensity of the membranes using the peak finder function, and normalizing it to the average background intensity.

## Acknowledgements

We thank SF Gabriel Krens, Jana Slováková, Shayan Shamipour, and Keisuke Sako for help with experimental design and data analysis, as well as the Heisenberg and Janovjak groups for discussion and feedback. We also thank Masazumi Tada and Daria Siekhaus for plasmids, and the Bioimaging and Zebrafish facilities of IST Austria for continuous support. This work was supported by an ERC Advanced Grant (MECSPEC) to C-PH. The Australian Regenerative Medicine Institute is supported by grants from the State Government of Victoria and the Australian Government. The EMBL Australia Partnership Laboratory (EMBL Australia) is supported by the National Collaborative Research Infrastructure Strategy (NCRIS) of the Australian Government.

## Additional information

### Funding

| Funder | Grant reference number | Author |
| --- | --- | --- |
| European Research Council | 742573 | Carl-Philipp Heisenberg |

The funders had no role in study design, data collection and interpretation, or the decision to submit the work for publication.

### Author contributions

Daniel Čapek, Formal analysis, Investigation, Visualization, Methodology, Writing—original draft, Writing—review and editing; Michael Smutny, Formal analysis, Writing—review and editing; Alexandra-Madelaine Tichy, Formal analysis; Maurizio Morri, Resources; Harald Janovjak, Conceptualization, Resources, Funding acquisition, Methodology, Writing—review and editing; Carl-Philipp Heisenberg, Conceptualization, Funding acquisition, Methodology, Writing—original draft, Project administration, Writing—review and editing

### Author ORCIDs

Daniel Čapek http://orcid.org/0000-0001-5199-9940
Michael Smutny https://orcid.org/0000-0002-5920-9090
Carl-Philipp Heisenberg https://orcid.org/0000-0002-0912-4566

### Ethics

Animal experimentation: Experiments involving zebrafish reported in this study included the creation and maintenance of transgenic mutant lines of zebrafish. In my role as Ethics Officer at IST Austria, I herewith confirm that the use of animals for experimental purposes within the scope of this study does not require the approval of the Ethics Committee at IST Austria. In line with EU directive 2010/

63/EU on the protection of animals used for scientific purposes, Austrian legislation requires that the use of animals for scientific purposes has to be approved by the respective unit at the Federal Ministry of Education, Science, and Research. This process includes an ethical evaluation of the planned experiments as part of the harm-benefit analysis. Beyond that no further approval by the Ethics Committee at IST Austria is required. The respective approval number that covers the experiments that were performed is 66.018/0010-WF/II/3b/2014. Yours sincerely, Dr. Verena Seiboth Ethics Officer

## Decision letter and Author response

Decision letter https://doi.org/10.7554/eLife.42093.031
Author response https://doi.org/10.7554/eLife.42093.032

## Additional files

### Supplementary files

• Transparent reporting form
DOI: https://doi.org/10.7554/eLife.42093.029

### Data availability

All data generated or analysed during this study are included in the manuscript and supporting files. Source data files have been provided for Figure 1—figure supplement 1, Figure 4—figure supplements 1, 3-5 and Figure 5—figure supplements 1-4.

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
