## [Decision Letter]

Thank you for submitting your article "Optogenetic Frizzled7 reveals a permissive role of non-canonical Wnt signaling in mesendoderm cell migration" for consideration by *eLife*. Your article has been reviewed by three peer reviewers, and the evaluation has been overseen by Didier Stainier as the Senior and Reviewing Editor. The reviewers have opted to remain anonymous.

The reviewers have discussed the reviews with one another and the Reviewing Editor has drafted this decision to help you prepare a revised submission.

In this manuscript, Capek and coworkers analyzed the requirement of non-canonical Wnt signaling in mesenchymal migration during zebrafish gastrulation. While it is well-established that Wnt/PCP signaling through the Frizzled receptor 7 (Fz7) controls epithelial polarized movement, it remains unclear whether this particular signaling system provides an instructive or a permissive role during mesenchyme migration. To address this question, the authors developed an optogenetic Fz7 construct (Opto-Fz7) and used it to study the impact of localized vs. global activation of the pathway in directing mesendodermal migration. The authors carefully quantify the polarized protrusions that drive ppl cell migration both in vitro and in vivo, and show that these are defective in Fz7 mutants; they further show that these defects can be rescued by uniform illumination (and thus activation) of Opto-Fz7. These data lead the authors to conclude that Fz7 plays only a permissive role, rather than an instructive or polarizing role.

To fully support the authors' claims and to strengthen the conclusions, the following points should be addressed:

1) Activity of Opto-Fz7a) The demonstration that Opto-Fz7 is active is based on the internalization of the receptor upon light exposure and changes in cell morphology. As this is the first report describing such a tool, it would be essential to directly demonstrate pathway activation with a more precise approach (e.g. changes in GFP-Dsh localization upon Opto-Fz7 activation).

b) To further validate the experiments, the authors should show the results for a Rhodopsin control in Figure 5—figure supplement 2.

c) As a control, the authors should assess how this tool affects Wnt signaling.

2) Distribution of Opto-Fz7a) The conclusions depend on the assumption that the distribution of injected Opto-Fz7 is uniform throughout the embryo, but only small tissue regions are shown in Figure 4 and Figure 4—figure supplement 1. To strengthen the conclusions, the authors should provide data on the distribution of Opto-Fz7-mKate2 throughout the embryo.

b) The authors state that they "found no evidence for a clearly recognizable polarized subcellular distribution of Fz7 or any other Wnt-Fz/PCP components in ppl cells (Figure 4B and not shown)". Please show the full data set or modify this claim.

c) Planar polarized turnover of PCP proteins is central to their function (see papers from Strutt), and a recent paper from the Kintner lab (Chien, Current Biology) shows that while Vangl2 has a polarized turnover and polarized activity, the protein is uniformly localized. While FRAP analysis to analyze the dynamics of the artificial Opto-Fz7 construct is beyond the scope of this manuscript, the authors should at least acknowledge this caveat in the text.

3) Cell migration induced by Opto-Fz7a) Video 4 shows a sudden movement of the cell aggregate, which is difficult to reconcile with a direct migration process. In this respect, it is unclear how a global activation of the illuminated cells in half of the tissue should result in a directed movement of the tissue towards the illuminated area. If cells are activated globally, why would they move towards the area of illumination? The authors should clarify the logic underlying these data or remove them.

b) In Figure 4G, please show single traces for Opto-Fz7 to facilitate the comparison with the highly variable Rhodopsin control. To strengthen the conclusions, a similar number of traces should be provided for both the Rhodopsin control and the OptoFz7 experiment (ideally 6 or more in both cases).

4) Data duplication

The data used to generate the graphs presented in Figure 1C, D seem to be reused in Figure 5D, E. The data presented in Figure 2C, D are reused in Figure 5G. The data presented in Figure 3B are reused in Figure 5B and Figure 5—figure supplement 2C. The image in Figure 1—figure supplement 1A is reused in Figure 5—figure supplement 3A. The authors should therefore restructure the figures to avoid this "data recycling" throughout the manuscript or at least clearly point out in the figure legends which data points are reused.

5) Manuscript organizationa) Please correct the following inaccuracies.Shindo and Wallingford 2010 is cited in a statement about epithelial cells, but that paper studies gastrula mesenchyme cells. Sepich et al., 2000 is cited, but that work on trilobite preceded identification of the locus as Vangl2. Jessen et al. 2002 should be included. Introduction, last paragraph: The statement that Wnt/PCP is required for directed migration of the prechordal plate should include "in zebrafish," as it is not required in *Xenopus* and there is no evidence for such a requirement in mice.

b) Many references are missing (e.g. Winklbauer et al., 2001, Gao et al., 2018, Marlow et al., 2002), and the bibliography contains several incorrect references (e.g. Ciruna et al., 2006 was published in Nature, not in Nature Cell Biology). Please correct this throughout the manuscript.

c) The supplementary source data files mostly show cumulative data, but for better transparency please organize the data by independent experiments rather than cumulatively.

d) Figure 5B: Please sort the data as in Figure 5D+E to enhance readability.

e) The authors might consider simply not abbreviating Rhodopsin, as the word is not used very often and "Rho" may be confusing given the tight links between PCP and Rho GTPases.

---

## [Author Response]

[…] To fully support the authors' claims and to strengthen the conclusions, the following points should be addressed:1) Activity of Opto-Fz7a) The demonstration that Opto-Fz7 is active is based on the internalization of the receptor upon light exposure and changes in cell morphology. As this is the first report describing such a tool, it would be essential to directly demonstrate pathway activation with a more precise approach (e.g. changes in GFP-Dsh localization upon Opto-Fz7 activation).

We have now co-expressed zebrafish Dvl2-GFP together with Opto-Fz7- mcherry and compared their localization in light and dark samples at the animal pole. We found that OptoFz7 recruited Dvl to the membrane, and that the membrane-to-cytosol ratio of Dvl decreased upon light exposure. This suggests that light activation of OptoFz7 triggers Fz7/Dvl endocytosis, previously shown to be required for Wnt/PCP signaling (Yu et al., 2007, Kim et al., 2008).

b) To further validate the experiments, the authors should show the results for a Rhodopsin control in Figure 5—figure supplement 2.

We have now added Rhodopsin controls to both the in vitro explant (uniform activation) and mutant embryo rescue (uniform activation) experiments.

c) As a control, the authors should assess how this tool affects Wnt signaling.

To rule out that canonical Wnt signaling is activated by Opto-Fz7 stimulation, we have now analyzed nuclear β-catenin localization as readout of canonical Wnt signaling in embryos exposed to either Opto-Fz7 or Wnt8 signaling (positive control) and compared them to un-injected wild type embryos (negative control). While Opto-Fz7-activated embryos, much like their wild type counterparts, displayed a higher intensity of nuclear β-catenin signal at the margin than at the animal pole, β-catenin localized to the nucleus both at the margin and animal pole in Wnt8 overexpressing embryos. This suggests that Opto-Fz7 activation does not trigger canonical Wnt signaling.

2) Distribution of Opto-Fz7a) The conclusions depend on the assumption that the distribution of injected Opto-Fz7 is uniform throughout the embryo, but only small tissue regions are shown in Figure 4 and Figure 4—figure supplement 1. To strengthen the conclusions, the authors should provide data on the distribution of Opto-Fz7-mKate2 throughout the embryo.

We have now expressed Opto-Fz7-mCherry in MZ*fz7a/b;gsc::*GFP-CAAX embryos, and analyzed its subcellular localization in different tissues of the gastrulating embryo. We found that Opto-Fz7-mCherry was uniformly distributed not only in prechordal plate cells (as shown in the original version of our manuscript) but also in epiblast, paraxial mesoderm, and notochord cells.

b) The authors state that they "found no evidence for a clearly recognizable polarized subcellular distribution of Fz7 or any other Wnt-Fz/PCP components in ppl cells (Figure 4B and not shown)". Please show the full data set or modify this claim.

We have now analyzed Wnt11-3x-mCherry, Pk1-venus, Dvl2-GFP, and Fz7a- mNeongreen subcellular distribution in prechordal plate cells, showing no evidence for a clearly recognizable polarized distribution.

c) Planar polarized turnover of PCP proteins is central to their function (see papers from Strutt), and a recent paper from the Kintner lab (Chien, Current Biology) shows that while Vangl2 has a polarized turnover and polarized activity, the protein is uniformly localized. While FRAP analysis to analyze the dynamics of the artificial Opto-Fz7 construct is beyond the scope of this manuscript, the authors should at least acknowledge this caveat in the text.

We have now acknowledged this caveat in the text.

3) Cell migration induced by Opto-Fz7a) Video 4 shows a sudden movement of the cell aggregate, which is difficult to reconcile with a direct migration process. In this respect, it is unclear how a global activation of the illuminated cells in half of the tissue should result in a directed movement of the tissue towards the illuminated area. If cells are activated globally, why would they move towards the area of illumination? The authors should clarify the logic underlying these data or remove them.

We have now repeated these experiments with an additional membrane marker and found that upon light activation cells on the activated side of the cluster are more protrusive than on the non-activated side. This suggests that cluster movement is caused by increased cell motility on the activated side.

b) In Figure 4G, please show single traces for Opto-Fz7 to facilitate the comparison with the highly variable Rhodopsin control. To strengthen the conclusions, a similar number of traces should be provided for both the Rhodopsin control and the OptoFz7 experiment (ideally 6 or more in both cases).

We have added more data, now showing six tracks per condition. These data are shown as mean with error bars in the main figure and as single tracks in a new supplementary figure.

4) Data duplicationThe data used to generate the graphs presented in Figure 1C, D seem to be reused in Figure 5D, E. The data presented in Figure 2C, D are reused in Figure 5G. The data presented in Figure 3B are reused in Figure 5B and Figure 5—figure supplement 2C. The image in Figure 1—figure supplement 1A is reused in Figure 5—figure supplement 3A. The authors should therefore restructure the figures to avoid this "data recycling" throughout the manuscript or at least clearly point out in the figure legends which data points are reused.

We repeated experiments to avoid data recycling where it was feasible (i.e. for the in situ-based experiments). Otherwise, we have now more clearly stated in the figure legends that results are compared to data already shown earlier.

5) Manuscript organizationa) Please correct the following inaccuracies.Shindo and Wallingford 2010 is cited in a statement about epithelial cells, but that paper studies gastrula mesenchyme cells. Sepich et al., 2000 is cited, but that work on trilobite preceded identification of the locus as Vangl2. Jessen et al. 2002 should be included. Introduction, last paragraph: The statement that Wnt/PCP is required for directed migration of the prechordal plate should include "in zebrafish," as it is not required in Xenopus and there is no evidence for such a requirement in mice.

We have now corrected this.

b) Many references are missing (e.g. Winklbauer et al., 2001, Gao et al., 2018, Marlow et al., 2002), and the bibliography contains several incorrect references (e.g. Ciruna et al., 2006 was published in Nature, not in Nature Cell Biology). Please correct this throughout the manuscript.

We have now corrected this.

c) The supplementary source data files mostly show cumulative data, but for better transparency please organize the data by independent experiments rather than cumulatively.

We have reorganized these data files.

d) Figure 5B: Please sort the data as in Figure 5D+E to enhance readability.

In Figure 5B, we show a gain-of-function experiment in wt and a rescue experiment in mutant, and we compare the treated groups to the respective untreated ones. In 5D and E, in contrast, we compare the rescued mutant under light condition to the wt, and the dark condition to the un- injected mutant. Since the data sets of the two experiments are different, we would prefer to keep these figures organized as they currently are.

e) The authors might consider simply not abbreviating Rhodopsin, as the word is not used very often and "Rho" may be confusing given the tight links between PCP and Rho GTPases.

This has been done.